# Effect of Culture Period and Stocking Density on Input Demand and Scale Economies of Milkfish (*Chanos chanos*) Polycultures with White Shrimp (*Penaeus indicus*)

**Wei-Tse Pai** [1], **Christian Schafferer** [2,*], **Jie-Min Lee** [3,*], **Li-Ming Ho** [4], **Yung-Hsiang Lu** [5], **Han-Chung Yang** [4] and **Chun-Yuan Yeh** [2]

[1] Department of Finance, National Changhua University of Education, Changhua City 50074, Taiwan; awaha0955@gmail.com
[2] Department of International Trade, Overseas Chinese University, Taichung City 40721, Taiwan; iune@ocu.edu.tw
[3] Department of Shipping and Transportation Management, National Kaohsiung University of Science and Technology, Kaohsiung City 81157, Taiwan
[4] Department of Marine Leisure Management, National Kaohsiung University of Science and Technology, Kaohsiung City 81157, Taiwan; lmhonkmu@gmail.com (L.-M.H.); hanchung@nkust.edu.tw (H.-C.Y.)
[5] Department of BioBusiness Management, National Chiayi University, Chiayi City 600355, Taiwan; yhlu@mail.ncyu.edu.tw
[*] Correspondence: chris@ocu.edu.tw (C.S.); jmlee866@yahoo.com.tw (J.-M.L.)

**Abstract:** Milkfish, *Chanos chanos*, is one of the major inland cultured fish species in Taiwan. Variations in land resources and climate have led to the application of two distinct culture practices of milkfish polycultures with white shrimp, *Penaeus indicus*. This study applies a translog cost function model to analyze the production scale economy and input demand price elasticity of four milkfish polyculture systems with two different culture periods (OWC and NOWC) and two different white shrimp–milkfish fry stocking ratios (low SMR: 10–55 fry/ha; high SMR: 56–100 fry/ha). The findings show that the four milkfish polyculture systems require different operational adjustments to increase production while reducing the average culture cost. More specifically, overwinter cultures (OWC) have economies of scale. Farmers may reduce the average cost by expanding the production scale. Non-overwinter polycultures (NOWC) with high SMR are at the stage of decreasing return to scale, meaning that gains in output of milkfish cannot reduce the average cost. In terms of input factor use, farmers of OWC systems with high SMR are sensitive to fluctuations in the fry price since fry constitutes the input factor exhibiting the highest own-price elasticity. Moreover, fry and feed of OWC households with high SMR have high levels of substitutability, whereas fry and other input exhibit substitutability in OWC systems with low SMR. In NOWC farming households with high SMR, fry and capital have substitutability. It is thus recommended to modify the input factor use according to the culture mode and the white shrimp–milkfish stocking density ratio. Moreover, the study found that NOWCs have considerably higher SMR than OWCs, which may lead to a deterioration of the water quality in NOWC fishponds and lower survival rates. It is thus recommended to reduce the SMR to 31:1 to achieve economies of scale in production and increase the survival rate of milkfish and white shrimp.

**Keywords:** *Chanos chanos*; polycultures; white shrimp–milkfish fry stocking ratios; economies of scale; price and cost elasticities

## 1. Introduction

Milkfish, *Chanos chanos*, is an important tropical fish in the Indo-Pacific region. In Taiwan, it is the second major inland cultured fish species. The main milkfish culture areas are in the southern part of the island with Chiayi, Tainan, and Kaohsiung as the top three regions. Variations in land resources and climate have led to different regional culture

practices [1]. In Taiwan, farmers add white shrimp, Penaeus indicus, to milkfish cultures. Polycultures of two or more compatible aquatic species result in a higher production compared with monocultures, mitigating losses caused by the relatively low fish price at peak harvest [2,3]. Past studies have also shown that milkfish polycultures with white shrimp increase the sustainability of production [4,5]. Waste produced by the main species in polycultures and feed residuals can be utilized by co-cultured species and turned into additional harvestable biomass [6].

Shrimp species in milkfish polycultures have beneficial effects on the water quality of ponds as well as adverse ecological effects [7,8]. Several studies concluded that increasing the quantity of white shrimp in milkfish polycultures has no significant effect on the survival rate of white shrimp [4,9]; however, excessive feed supplied as a result of higher stocking densities of milkfish fry may lead to a significant deterioration of the water quality in fishponds, thus negatively affecting the growth and survival rate of shrimp species [10,11]. As such, farming households of milkfish polycultures with shrimp species are advised to control the shrimp stocking density to enhance input factor management and increase profit [1].

Previous studies on milkfish cultures have been mostly concerned with the growth and survival rate of milkfish as well as the ecological conditions of fishponds in terms of water quality [4,8,12–16]. There have, however, been fewer studies regarding the production scale economy of milkfish. Lee et al. (2020), for example, analyzed the scale economies and factor utilization in terms of milkfish fry stocking density and fry size. The study found that the cultivation of small fry (2–3 inches) with high or low stocking densities had better production scale economy than cultures with large fry (≥4 inches) Moreover, high-density stocking with small fry exhibited a higher fry price elasticity. The authors thus suggested that farming households could modify the fry stocking density and fry size to increase the overall production output while reducing cost.

Apart from modifying operational characteristics to increase productivity, other studies have shown that the production scale economy of milkfish cultures is also determined by the farmer's culture management ability, his or her age, experience, and educational attainment [16,17]. Mohan et al. for instance, in their study on the technical inefficiency among farmers operating at different intensity levels, found that additional educational opportunities could substantially increase the efficiency of fish production without major new investments [18].

This study addresses two research questions. First, it estimates the relationship between production scale economies and input factor use of four different systems of milkfish polycultures with white shrimp. Second, it investigates whether the different polycultures exhibit cost complementarity/substitutability. More specifically, by applying a translog cost function model, the production scale economy and input–input demand service condition of the elasticity of four milkfish polyculture systems with two different culture periods (NOWC: 7–8 months; OWC: 12 months) and two different white shrimp-milkfish fry stocking ratios (low: 10–55 fry/ha; high: 56–100 fry/ha) are analyzed. Culture production and input factor data of 120 farming households for the years 2018 to 2019 are used in the analysis.

## 2. Methodology and Data

### 2.1. Study Areas and Culture Mode

According to the culture period, two distinct culture practices are common in Taiwan. In the counties of Chiayi and Tainan, stocking and harvesting are both performed in the same year. In our study, this culture system is referred to as non-overwinter culture (NOWC). In Kaohsiung, the grow-out period is extended into the winter season. Overwinter cultures (OWC) are common in the Kaohsiung area because of the widespread use of deep-water ponds and the higher average temperature in winter due to the more southerly latitude. Deep-water ponds provide a more stable environment than conventional shallow water ponds. They allow farmers to culture fish at much higher stocking density and to

extend the culture period. The average culture period for OWC systems is twelve months, whereas seven to eight months are common for NOWC farming.

The production output of the counties of Tainan and Kaohsiung accounts for about 85% of Taiwan's annual milkfish production (Figure 1). As both areas are also the most representative in terms of the two distinct culture practices, OWCOWC and NOWC, farming households in the two counties are the subjects of this study [19]. The milkfish farmers were clustered into four groups according to culture practice and the white shrimp–milkfish fry stocking density ratio (SMR). In total, 60 households (260 fishponds) in each of the two counties were included in the study.

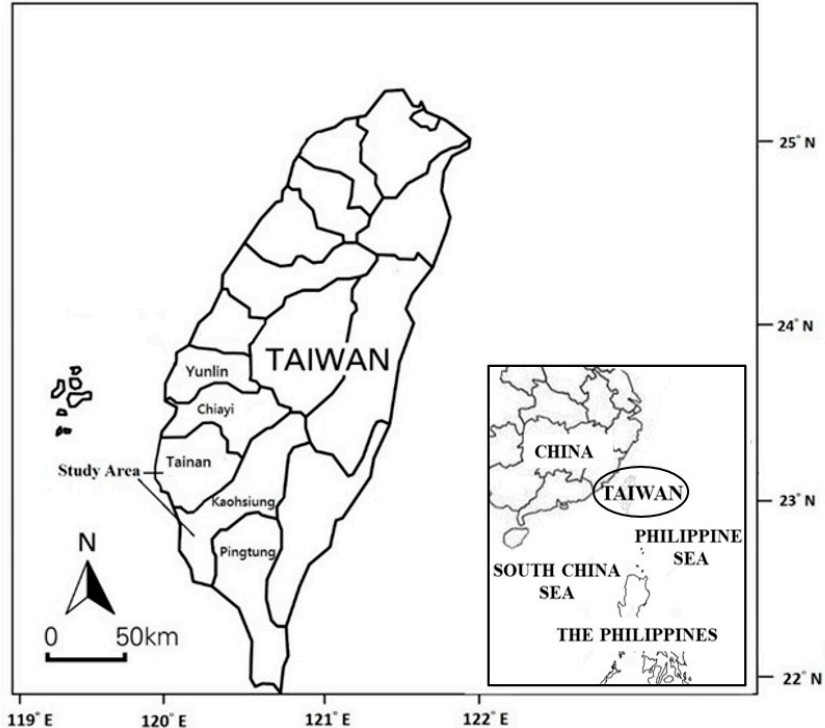

**Figure 1.** Geographical locations of the milkfish farms in Tainan County and Kaohsiung County, Taiwan.

The adult fish culture work includes pond preparation, stocking, feeding, pond management, and harvesting. Fishpond preparation for NOWCs is carried out between January and March, and for OWCs, it is carried out from March to May. It includes draining and pond solarization, conditioning and repairing dikes, water intake and drainage facilities, as well as removing pests and waste fish.

The farmers select the milkfish fry stocking size and stocking density according to the expected harvest time and fry price. NOWC farming has a shorter culture period. As such, milkfish fry are released either in the middle or at the end of April. White shrimp are added during the culture process. The average stocking density of milkfish fry is kept below 10,000 fry/ha, and the average stocking density of white shrimp fry about 500,000 fry/ha. The white shrimp fry used in the polycultures can be released in 2–4 stages.

The overwinter culture has a longer culture period. Milkfish fry are released in June. The average stocking density of milkfish fry is 20,000 fry/ha, and about 800,000 fry/ha for white shrimp fry, which are released in stages.

As for the pond management, the dissolved oxygen in the fishpond must be measured constantly. When the dissolved oxygen decreases, the waterwheel must be actuated to prevent the fish school dying from oxygen deficiency. In terms of feed supply, the automatic dispenser supplies artificial feed to save labor costs. White shrimp in polycultures eat the leftover feed and fish excrement, thus cleaning the water. It is required to regularly check whether the cultured milkfish is affected by diseases or insects.

The harvesting period of NOWCs is between September and November. After a culture period of three months, white shrimp are harvested between July and December. In the case of OWCs, milkfish are harvested mainly in December, January, and February, and white shrimp are harvested between September and February.

## 2.2. Empirical Model

This study applied translog cost function modelling to analyze the output and cost input data of milkfish and white shrimp polycultures [20]. The translog cost function has been widely used in various studies to investigate production cost structures and production input factors [21–24]. More specifically, this study used a translog cost function of milkfish polycultures with white shrimp, five production inputs (labor, fry, capital, feed and other miscellaneous), and two outputs (milkfish and white shrimp). The translog cost function of milkfish polycultures with white shrimp is specified as per Equation (1).

$$
\ln C = \alpha_0 + \sum_k \alpha_k \ln Y_k + \frac{1}{2} \sum_k \sum_l \alpha_{kl} \ln Y_k \ln Y_l + \sum_i \beta_i \ln P_i + \frac{1}{2} \sum_i \sum_j \beta_{ij} \ln P_i \times \ln P_j + \sum_k \sum_i \gamma_{ki} \ln Y_k \times \ln P_i
$$
$$
+ \eta_0 \times \text{Ratio} + \sum_k \eta_k \text{Ratio} \times \ln Y_k \qquad \forall i, j = S, F, K, L, O \ k = m, s \tag{1}
$$

where C is the total cost of production; Ym and Ys are the vectors of output of milkfish and white shrimp, respectively, and Pi is the vector of input factor price. The five production factors are fry (S), feed (F), capital (K), labor (L), and other miscellaneous (O). $\alpha_0$, $\alpha_k$, $\alpha_{kl}$, $\beta_i$, $\beta_{ij}$, $\gamma_{kl}$, $\eta_0$ and $\eta_k$ are the estimated parameters. The total operating cost of production corresponds to the sum of labor cost, fry cost, fund cost, and other costs.

According to Shephard's lemma, if the factor price is differentiated by Equation (1), the cost share (Si) in Equation (2) can be obtained.

$$
\partial \ln C \Big/ \partial \ln P_i = S_i = \beta_i + \sum_j \beta_{ij} \ln P_j + \sum_k \gamma_{ki} \ln Y_k \tag{2}
$$

To correspond to a well-behaved production function, the production function must meet the input factor price homogeneity of degree one and symmetry requirements. The constraints include

$$
\sum_i \beta_i = 1, \ \sum_i \beta_{ij} = \sum_j \beta_{ji} = 0, \ \sum_i \gamma_{ki} = 0, \ \beta_{ij} = \beta_{ji} \tag{3}
$$

Imposing symmetry and homogeneity by using parameter constraints, the cost function (1), and cost-share (2) are jointly estimated using SUR methods proposed by Zellner (1962) [25]. As mentioned, only *n*-1 factor shares equations are linearly independent.

## 2.3. Economies of Scale and Input Demand Price Elasticity

To further analyze the cost structure characteristics of milkfish, the economies of scale and input demand price elasticity indexes can be calculated by the estimated parameters.

Economies of scale are said to exist if long-term average costs decline as output increases [26]. In a multiproduct setting, the overall economies of scale (OSE) can be measured as $1/\sum \varepsilon_{CY_k}$, with $\varepsilon_{CY_k}$ being the elasticity of total cost with respect to output $Y_K$.

The specific scale economies (PSE) are the inverse of the elasticity of the cost ($\varepsilon_{CY_k}$). $\varepsilon_{CY_k}$ is defined as the percentage change in total cost (C) caused by the percentage change in quantity ($Y_k$), which can be calculated by differentiating the natural log of total cost (lnC) with respect to the natural log of quantity ($lnY_k$) using the following equation:

$$
\varepsilon_{CY_k} = \frac{\partial \ln C}{\partial \ln Y_k} = \alpha_k + \sum_l \alpha_{kl} \ln Y_l + \sum_i \gamma_{ki} \ln P_i + \eta_k \text{Ratio} \tag{4}
$$

Values of scale economies larger than, equal to, or smaller than one imply increasing, constant, or decreasing returns to scale, respectively.

When the specific scale economies' (PSE) value of milkfish is 1, the farmer's output has reached the minimum efficient scale (MES), and the average cost of farmer has reached the lowest level. To assess the MES, the linear relationship of milkfish production output, the output interaction of fry stocking density ratio of milkfish to white shrimp, and the specific scale economies, should be estimated as shown in Equation (5):

$$PSE_i = \rho_0 + \rho_1 Y_i + \rho_2 Ratio_i + \rho_3 Y_i \times Ratio_i + \varepsilon_i \tag{5}$$

where $PSE_i$ is the specific scale economies, $Y_i$ is the production quantity of the i farmer, $Ratio_i$ is the fry stocking density ratio of milkfish to white shrimp, and $\varepsilon_i$ is an error term. $\rho_0$, $\rho_1$, $\rho_2$ and $\rho_3$ are the estimated parameters. To minimize the effect of outliers, we performed the estimation using a robust linear regression model rather than an ordinary least square model.

The input demand price elasticity is defined as the effect of input factor price change on the change in input demand when the other conditions remain constant. If the absolute value of the price elasticity of input demand is greater than 1, input demand is termed price elastic. If it is equal to 1, input demand is unit price elastic. If it is less than 1, input demand is price inelastic. The own-price elasticities of inputs demand are used to measure the demand response of input i with respect to changes in the price of input i, as expressed below:

$$\eta_{ii} = \frac{\beta_{ii}}{s_i} + s_i - 1 \tag{6}$$

To further understand the substitutability and complementarity between input factors, the Allen partial elasticities (ASE) can be calculated, (Allen partial elasticities, ASE) which is a net or Hicksian elasticity. Allen partial elasticities of substitution between factors i and j are calculated as follow:

$$\delta_{ij} = \frac{\beta_{ij}}{s_i s_j} + 1 \text{ for } i \neq j \tag{7}$$

### 2.4. Determinants of Overall Scale Economies

According to the main factors affecting milkfish farming, in terms of overall scale economies (OSE), the basic model affecting milkfish OSE is established as follows

$$OSE_i = \theta_0 + \theta_1 Area_i + \theta_2 Time_i + \theta_3 Depth_i + \theta_4 Fresh_i + \theta_5 Density_{mi} + \theta_6 Density_{si} + \theta_7 Size_i + \theta_8 Age_i + \theta_9 Exp_i + \theta_{10} Edu_i + \theta_{11} Winter_i + u_i \tag{8}$$

where the explained variable OSE is the farming households' overall scale economy; Area is the culture area of the farm investigated; Time denotes the culture period per production cycle; Depth is the pond's water depth; Fresh is the water source of fresh water; $Density_m$ and $Density_s$ are the fry stocking density of milkfish and white shrimp, respectively; Size denotes the size of the milkfish fry stocked in the farm per production cycle; Age is the farmer's age; Exp is the number of years the farmer has been engaged professionally in fish farming; Edu is the famer's education at college level or above; Winter refers to the overwinter culture system (OWC); $u_i$ is the random interference term; $\theta_0$ is the constant term.

### 2.5. Data Sources and Variables Definitions

This study investigated overwinter adult milkfish farmers in the county of Kaohsiung (OWC) and non-overwinter adult milkfish farmers in the county of Tainan (NOWC). Data regarding the culture operation of the farming households, as well as personal details of the head farmers, were investigated. Convenience sampling was used. The local fishermen's associations provided the sample data. Five fishermen's associations were selected in Kaohsiung and Tainan, respectively. That is, a total of ten fishermen's associations were investigated. Professionally trained interviewers surveyed the members of the selected

fishermen's associations using structured questionnaires. In order to control for potential sampling bias and to check for unusual as well as inconsistent responses, in-depth interviews were conducted with two representatives of the milkfish industry (feed and fry suppliers): two senior fishermen, and one scholar. The final sample included the responses of 60 farming households (260 fishponds) in each of the two observed geographical regions, accounting for about 4% of the 2931 farming households in Kaohsiung and Tainan. Each farmer completed the survey in the years 2018 and 2019. As such, a total of 240 questionnaires were included in the study.

The culture operation and personal data of the observed farming households include the farmer's age, experience, and educational status, as well data regarding the culture pond depth, water source, and culture time. The biological data include the stocking density and stocking size of milkfish fry and white shrimp fry. The cost data is the outlay of production cost per hectare, which includes labor cost, fry cost, capital cost, feed cost, and other costs. The labor cost comprises the costs of family workers, workers, and temporary workers. The fry cost consists of the purchase costs of milkfish fry and white shrimp fry. The capital cost is mainly the equipment depreciation expenses. The equipment capital investment includes the costs of fishing rafts, watermills, water wells, generators, and culture huts. The feed cost includes feed cost and fertilizer cost. The other costs include water and electricity expenses, fishpond and equipment maintenance costs, a loan cost, and drug and insurance expenses.

This study used the cost share model of translog total cost and input factor for empirical analysis. The total cost, output, input factor price, cost outlay share, as well as variable definitions, are shown in Table 1. The total production cost (C) is the sum of labor cost, fry cost, fund cost, feed cost, and other costs, and the unit is NTD/ha. The output (Y) includes the outputs of milkfish and white shrimp (kg/ha). The production input factor price includes the price of labor ($P_L$), the total cost of family workers, workers, and temporary workers, and is divided by the culture area (NTD/ha). The fry price ($P_S$) is calculated according to the purchase prices of milkfish and white shrimp fry, and the weighted average price is calculated according to the ratio of buying expenses. The capital price ($P_K$) is calculated by dividing the equipment depreciation expense by the culture area (NTD/ha). The equipment consists of fishing rafts, watermills, water pumps, generators, the water quality, bottom soil measuring equipment, and farmhouses. The feed price ($P_F$) is calculated by dividing the total feed and fertilizer cost outlay by the weight of feed and fertilizer (NTD/kg). The other factor price ($P_o$) is calculated by dividing the sum of water and electricity expenses, fishpond and equipment maintenance costs, a loan cost, and drug and insurance expenses by the culture area (NTD/ha).

The milkfish farmers are clustered into four groups according to culture practice, OWC and NOWC, and the white shrimp–milkfish fry stocking density ratio (SMR). That is, (I) overwinter culture of milkfish (OWC) with low SMR (stocking density ratio 10–55); (II) overwinter culture of milkfish (OWC) with high SMR (stocking density ratio 56–100); (III) non-overwinter culture of milkfish (NOWC) with low SMR (stocking density ratio 10–55); (IV) non-overwinter culture of milkfish (NOWC) with high SMR (stocking density ratio 56–100). In polycultures with low SMR the quantity of white shrimp cultured with one individual of milkfish ranges between 10 and 55 fry, whereas the quantity of shrimp fry per milkfish is higher in the case of polycultures with high SMR.

A translog cost function model is used to estimate the cost function parameters, and economy of scale indexes of the four different culture systems. Moreover, the impact of operational characteristics on production scale economies values is analyzed.

**Table 1.** Description of the variables in Translog cost and scale economies determinants.

| Variable | Symbol | Description |
|---|---|---|
| | Cost variables | |
| Total production cost (NTD/ha) | C | Total cost of milkfish polyculture white shrimp produced |
| Cost share of inputs | $S_L$, $S_S$, $S_K$, $S_F$, $S_O$ | Represents cost share of input labor ($S_L$), fry ($S_S$), capital ($S_K$), feed ($S_F$), and other miscellaneous production inputs ($S_O$) |
| Output variables | | |
| Output of milkfish (kg/ha) | $Y_m$ | Total quantity of milkfish produced |
| Output of white shrimp(kg/ha) | $Y_s$ | Total quantity of white shrimp produced |
| | Input price variables | |
| Fry price (NTD/fry) | $P_S$ | The weighted average of milkfish and white shrimp fry buying price according to the ratio of purchase outlay |
| Capital price (NTD/ha) | $P_K$ | The equipment depreciation expense divided by the culture area |
| Feed price (NTD/kg) | $P_F$ | The total cost outlay for feed and fertilizer divided by the weight of feed and fertilizer |
| Labor price (NTD/ha) | $P_L$ | The total cost of family workers, workers, and casual laborers divided by the culture area |
| Other factor price (NTD/ha) | $P_o$ | The sum of water and electricity expenses, fishpond and equipment maintenance costs, loan cost, and drug and insurance expenses divided by the culture area |
| | Farm- and farmer characteristics affecting scale economics | |
| Overall scale economies | OSE | Estimates based on the estimated coefficients from the Translog cost model |
| Culture area (hectare) | Area | Represents the culture area of the farm |
| Culture time (months) | Time | Represents the culture period of the farm per production cycle |
| Water depth (meter) | Depth | Represents the pond's water depth |
| Water source (dummy) | Fresh | Status of the fish farmer water use. '1' indicates the use of fresh water, otherwise '0' |
| Milkfish fry stocking density (fry/ha) | $Density_m$ | Milkfish fry stocked per production cycle |
| White shrimp fry stocking density (fry/ha) | $Density_s$ | White shrimp fry stocked per production cycle |
| Milkfish fry stocking size (inches/fry) | Size | Milkfish fry stocked per production cycle |
| Age of the farmer (years) | Age | Represents the age of fish farmer |
| Experience of the farmer (years) | Exp | Represents the number of years the farmer spent in fish farming |
| Gender of the farmer (dummy) | Male | '1' indicates male, otherwise '0' |
| Education of the farmer (dummy) | Edu | '1' college or above, otherwise '0' |
| White shrimp–milkfish fry stocking density ratio | Ratio | White shrimp fry stocking density divided by milkfish fry stocking density |

## 3. Results

### 3.1. Summary Descriptive Statistics

Table 2 shows the descriptive statistics of the four polyculture systems. Overwinter households (OWC) mainly comprise low SMR polycultures, whereas high SMR cultures are more common in non-overwinter farming (NOWC).

**Table 2.** Descriptive data of the observed farming household clusters.

| Culture Mode | Overwinter (OWC) | | | Non-Overwinter (NOWC) | | | |
|---|---|---|---|---|---|---|---|
| **High White Shrimp–Milkfish Fry Stocking Density Ratio (SMR)** | **Low** | **High** | **Total** | **Low** | **High** | **Total** | **F Value [a]** |
| No. of samples | 72 | 48 | 120 | 50 | 70 | 120 | |
| Cost variables | | | | | | | |
| Total cost (NTD/ha) | 1,129,114 ± 630,266 | 920,667 ± 406,069 | 1,016,832 ± 558,433 | 528,773 ± 107,597 | 567,967 ± 143,316 | 551,636 ± 130,557 | 6.64 ** |
| Fry cost share (%) | 7.72 ± 3.07 | 10.05 ± 6.01 | 8.78 ± 4.65 | 9.05 ± 4.27 | 12.06 ± 6.00 | 10.86 ± 5.62 | 0.78 |
| Feed cost share (%) | 52.53 ± 15.41 | 46.36 ± 6.36 | 51.79 ± 12.53 | 31.25 ± 8.02 | 35.60 ± 10.97 | 33.86 ± 10.09 | 6.17 ** |
| Other cost share (%) | 15.65 ± 7.68 | 17.42 ± 3.81 | 16.73% ± 6.35 | 30.66 ± 9.75 | 24.97 ± 10.48 | 27.24 ± 10.59 | 8.06 ** |
| Labor cost share (%) | 21.63 ± 10.89 | 22.85 ± 8.68 | 22.69 ± 10.01 | 24.30 ± 7.67 | 21.89 ± 8.14 | 22.85 ± 8.04 | 0.69 |
| Capital cost share (%) | 2.47 ± 1.21 | 3.31 ± 1.10 | 2.84 ± 1.22 | 4.74 ± 1.52 | 5.48 ± 1.65 | 5.19 ± 1.62 | 0.12 |
| Output variables | | | | | | | |
| Output of milkfish (kg/ha) | 17,310 ± 12,556 | 11,376 ± 3640 | 14,936 ± 10,351 | 5396 ± 1586 | 6031 ± 1525 | 5767 ± 1576 | 12.69 ** |
| Output of white shrimp(kg/ha) | 924 ± 677 | 1268 ± 654 | 1062 ± 697 | 1085 ± 831 | 1271 ± 1349 | 1194 ± 1161 | 0.91 |
| Input price variables | | | | | | | |
| Fry price (NTD/fry) | 2.40 ± 0.73 | 2.21 ± 0.90 | 2.32 ± 0.80 | 4.43 ± 2.50 | 3.34 ± 2.55 | 3.79 ± 2.57 | 4.8 ** |
| Feed price (NTD/kg) | 15.18 ± 14.15 | 14.43 ± 12.60 | 14.88 ± 13.48 | 14.81 ± 113.18 | 14.59 ± 12.16 | 14.68 ± 12.46 | 1.54 |
| Other price (NTD/ha) | 95,666 ± 74,790 | 120,740 ± 12,2759 | 105,696 ± 97,353 | 96,220 ± 51,949 | 93,175 ± 83,565 | 94,444 ± 71,851 | 0.01 |
| Labor price (NTD/ha) | 105,370 ± 279,330 | 141,583 ± 361,550 | 119,855 ± 314,162 | 84,525 ± 63,615 | 93,913 ± 11,4975 | 90,001 ± 96,710 | 2.82 * |
| Capital prices (NTD/ha) | 17,626 ± 19,814 | 24,868 ± 26,063 | 20,523 ± 22,709 | 14,873 ± 7731 | 21,294 ± 20,856 | 18,618 ± 16,938 | 1.09 |
| Farm and farmer characteristics | | | | | | | |
| Culture area (ha) | 3.01 ± 1.43 | 2.18 ± 1.57 | 2.68 ± 1.50 | 2.10 ± 1.33 | 3.11 ± 4.59 | 2.69 ± 3.63 | 4.34 ** |
| Culture period (month) | 11.95 ± 1.86 | 10.22 ± 1.40 | 11.26 ± 1.90 | 8.04 ± 1.16 | 7.68 ± 1.61 | 7.83 ± 1.45 | 12.47 ** |
| Water depth (cm) | 4.31 ± 0.73 | 3.39 ± 0.77 | 3.94 ± 0.89 | 3.83 ± 0.67 | 3.52 ± 1.01 | 3.65 ± 0.89 | 9.56 ** |
| Water source (fresh water) (%) | 61.11 | 87.5 | 71.66 | 94.00 | 91.43 | 92.5 | |
| Milkfish fry stocking density (fry/ha) | 27,455 ± 18,202 | 17,413 ± 8160 | 23,304 ± 15,546 | 7406 ± 1752 | 8304 ± 1760 | 7406 ± 1751 | 16.3 ** |
| White shrimp fry stocking density(fry/ha) | 665,484 ± 563,041 | 1,189,715 ± 479,869 | 882,407 ± 588,260 | 244,282 ± 116,174 | 632,193 ± 210,358 | 432,239 ± 257,034 | 0.4 |
| Survival rate of milkfish | 84.8% ± 5.3% | 87.1% ± 4.5% | 85.5% ± 5.07% | 92.5% ± 5.9% | 90.6% ± 6.0% | 91.4% ± 5.93% | 6.94 ** |
| Survival rate of white shrimp | 9.5% ± 0.5% | 7.1% ± 0.5% | 8.2% ± 0.5% | 18.8%V0.9% | 9.9% ± 0.6% | 13.3% ± 0.7% | 5.51 ** |
| Milkfish fry stocking size (inches/fry) | 2.99 ± 1.33 | 3.42 ± 1.08 | 3.16 ± 1.25 | 3.34 ± 1.22 | 3.21 ± 1.58 | 3.26 ± 1.44 | 3.62 * |
| Age of household head (years) | 60.97 ± 11.59 | 62.62 ± 10.75 | 61.63 ± 11.25 | 59.32 ± 14.20 | 56.81 ± 14.25 | 57.85 ± 14.23 | 1.66 |
| Experience of household head (years) | 26.27 ± 10.32 | 20.62 ± 7.95 | 24.01 ± 9.81 | 28.14 ± 7.63 | 29.71 ± 9.78 | 29.05 ± 8.95 | 9.14 ** |
| Gender of the farmer (male %) | 95.83% | 91.67% | 94.16% | 92.00% | 85.71% | 88.33% | |
| Education of the college and above (%) | 13.89% | 12.50% | 13.33% | 16.00% | 30.00% | 24.16% | |
| White shrimp–milkfish fry stocking density ratio | 24 ± 14 | 71 ± 11 | 43 ± 26 | 33 ± 14 | 75 ± 16 | 53 ± 25 | 10.01 ** |

Note: 1 USD = 30.08 NTD. Values are expressed as mean ± SD. Abbreviations: USD, United States dollar; NTD, new Taiwan dollar; ha, hectare. [a] Conducting a test of null hypothesis that the combination means (expressed in mean vectors) caused by two factors of culture mode and the white shrim–milkfish fry stocking density ratio (SMR) are not statistically different. *, ** indicates statistical significance at 10% or 5%, respectively.

In terms of milkfish production output, OWC polycultures with low SMR have the highest average output of milkfish per hectare at 17,310 kg, and NOWC farmers with low SMR have the lowest at 5396 kg. Average production of white shrimp in high SMR polycultures ranges between 1268 kg (OWC) and 1271 kg (NOWC) per hectare. On average, the output of low SMR cultures has been slightly lower.

In terms of total cost, OWC systems with low SMR have the highest total cost per hectare at NTD 1,129,114, and NOWCs have the lowest at NTD 528,773.

Moreover, the feed cost accounts for the largest share of the total farming expenses. In OWCs with low SMR, it amounts to 52.53% of the total cost. Apart from the feed cost, the labor cost, and other costs, are significant cost factors. More specifically, the labor cost of OWC and NOWC farms ranges between 21.63 and 24.3% of the total cost. Other costs are the second largest cost factor of NOWC farming, reaching 30.66% of the overall cost in low SMR systems. The capital cost takes up the smallest proportion of the total culture cost, ranging between 2.47% and 5.48%.

As for the farmer and culture operation characteristics, OWCs with low SMR cover the largest average culture area (3.01 ha), have the longest average culture time (11.95 months), and the deepest culture ponds (4.31 m). The culture water source for OWC and NOWC systems is mainly brackish water. In terms of milkfish stocking density, OWCs with low SMR have the highest average stocking density at 27,455 fry/ha. NOWCs have considerably lower stocking densities, ranging between 7406 and 8304 fry/ha. In terms of the quantity of white shrimp, OWCs have the largest quantity of released white shrimp at 1,189,715 fry/ha with a maximum shrimp-milkfish ratio of 71.

### 3.2. Canonical Discrimant Function Analysis

A canonical discriminant function analysis was performed to determine whether the four polyculture systems are significantly different from each other in terms of cost structure and household characteristics. The corresponding means of the discriminant functions Can1 to Can3 and Can4 to Can6 for the four systems are shown in Supplementary Table S1 and Supplementary Table S2. In the discriminant analysis of cost input and household characteristics, the approximated F values for the discriminant functions was significant at the 0.05 level. The high Eigenvalue indicates that the functions differentiate the groups. In terms of cost inputs, feeding cost, and labor cost, (Can1) significantly discriminates between OWCs (NOWCs) with different SMRs. In terms of household characteristics, the two variables' culture period and white shrimp–milkfish ratio (Can4) significantly discriminate between groups.

### 3.3. Parameter Estimation

The cost function model parameters of the four clusters are estimated under the symmetry, homogeneity, and added-up constraints. In the estimation, the overwinter and non-overwinter farming households are subdivided into four clusters according to low and high white shrimp–milkfish ratios. Moreover, bootstrapping with 1000 iterations is performed to evaluate the estimation errors. The Lagrange multiplier test rejects the null hypothesis of no heteroscedasticity.

The parameter estimation results are shown in Supplementary Table S3. The estimated parameter results approximately meet the theoretical requirement of the cost function. That is, among the estimated parameters of the estimated total cost and cost share model, the Chi-square test value reaches statistical significance, implying that all the estimated parameters are not 0. As such, the model has predictive ability. As the adjusted R-square of the estimated total cost and cost share model ranges between 0.987 and 0.999, it may be assumed that the model fits the observed data.

### 3.4. Input Demand Price Elasticity Estimation

Table 3 shows the own and input demand price elasticity estimation results of the four clusters. When the absolute value of the own price elasticity of input demand is less than 1,

input demand is termed price inelastic. The estimated own-price elasticities of labor, fry, capital, feed, and other input demands of the four culture systems are lower than 1. As such, the five input demands of OWC and NOWC farms with different SMR lack price elasticity. The input factor fry of OWCs with high SMR has the highest own-price elasticity at 0.935, implying that the farmers are sensitive to fry price fluctuations.

**Table 3.** Estimated own-price and Allen partial substitution elasticities of the observed farming household clusters.

| | Overwinter (OWC) | | | | | | | | | |
| --- | --- | --- | --- | --- | --- | --- | --- | --- | --- | --- |
| | Low White Shrimp–Milkfish Ratio | | | | | High White Shrimp–Milkfish Ratio | | | | |
| | Price for | | | | | Price for | | | | |
| Demand for | Labor | Fry | Capital | Feed | Others | Labor | Fry | Capital | Feed | Others |
| Labor | −0.076 (0.046) | 0.343 (0.405) | −0.245 (0.677) | 0.132 (0.082) | 0.603 (0.127) ** | −0.535 (0.212) ** | 1.789 (1.035) | 1.542 (1.049) | 0.664 (0.234) ** | 0.573 (0.206) ** |
| Fry | | −0.465 (0.131) ** | 0.577 (0.960) | 0.277 (0.233) | 1.121 (0.168) ** | | −0.935 (0.195) ** | 0.045 (1.844) | 1.214 (0.336) ** | 0.625 (0.202) |
| Capital | | | −0.087 (0.190) | 0.113 (0.152) | 0.717 (0.195) ** | | | 0.279 (0.375) | −1.275 (0.705) | 0.550 (0.715) |
| Feed | | | | −0.121 (0.027) ** | 0.717 (0.025) ** | | | | −0.363 (0.071) ** (0.517) ** | 0.889 (0.113) ** |
| Other miscellaneous | | | | | −0.284 (0.037) ** | | | | | −0.357 (0.068) ** |
| | Non-overwinter (NOWC) | | | | | | | | | |
| | Low White Shrimp–Milkfish Ratio | | | | | High White Shrimp–Milkfish Ratio | | | | |
| | Price for | | | | | Price for | | | | |
| Demand for | Labor | Fry | Capital | Feed | Others | Labor | Fry | Capital | Feed | Others |
| Labor | −0.026 (0.030) | −0.539 (0.354) | 0.039 (0.605) | 0.401 (0.215) | −0.078 (0.163) | −0.036 (0.039) | 0.277 (0.206) | −1.034 (0.411) ** | 0.551 (0.173) ** | 0.107 (0.105) |
| Fry | | −0.125 (0.242) | 0.857 (2.902) | 0.215 (1.417) | 0.520 (0.560) | | −0.550 (0.079) ** | 1.338 (0.623) ** | 1.027 (0.234) ** | 0.349 (0.203) |
| Capital | | | −0.337 (0.188) | 0.299 (2.014) | 0.582 (1.164) | | | −0.022 (0.122) | 0.893 (0.544) | 0.043 (0.245) |
| Feed | | | | −0.409 (0.157) ** | 0.622 (0.523) | | | | −0.527 (0.100) ** | 1.016 (0.243) ** |
| Other miscellaneous | | | | | −0.206 (0.090) ** | | | | | −0.243 (0.029) ** |

\*\* indicates statistical significance at 5%. Standard errors, calculated using the delta method, are in parentheses.

Table 3 shows the estimation result of the Allen partial elasticities of substitution. Values of the partial elasticities of substitution above zero suggest substitutional relations between factors, whereas values below zero indicate complementarity of factors. In OWCs with low SMR, fry and other input show relatively high substitutability. The partial elasticities of substitution equal 1.121. The input factors fry *I* and feed of polycultures with high SMR exhibit high levels of substitutability. The partial elasticities of substitution in this case reach 1.214. In NOWCs, fry and capital have the highest level of substitutability. More specifically, polycultures with high SMR exhibit the highest value of partial elasticity of substitution at 1.338. The input factors, labor and capital, of NOWCs with high white SMR show relatively high levels of complementarity with the partial elasticities of substitution reaching −1.034.

### 3.5. Scale Economies Estimation

Table 4 shows the estimation result of the overall scale economies' value of OWC and NOWC farming households. The proportion of the overall SE of OWC farmers using low SMR within 0.9–1.0 is the highest at 38%, next to 32.9% within 1.0–1.1. The proportion of overall SE of the OWC farmers using high SMR within 0.9–1.0 is the highest at 27.1%, followed by 22.9% within 1.0–1.1. The average overall SE values of the OWC farmers are

slightly greater than 1, representing the stage of increasing return to scale, meaning that additional gains in output contribute to reducing the average cost.

**Table 4.** Distribution of overall scale economies scores of low and high SMR in OWCs and NOWCs.

| Culture Mode | Overwinter (OWC) | | Non-Overwinter (NOWC) | |
|---|---|---|---|---|
| White Shrimp–Milkfish Fry Stocking Density Ratio (SMR) | Low | High | Low | High |
| Range of Overall Scale Economies (OSE) | % of Farmers in OSE Interval | % of Farmers in OSE Interval | % of Farmers in OSE Interval | % of Farmers in OSE Interval |
| <0.7 | | | 2 | 17.4 |
| 0.7–0.8 | | 8.3 | 3.9 | 63.8 |
| 0.8–0.9 | 11.4 | 10.4 | 7.8 | 15.9 |
| 0.9–1.0 | 38.0 | 27.1 | 37.3 | 0 |
| 1.0–1.1 | 32.9 | 22.9 | 45.1 | 1.4 |
| 1.1–1.2 | 12.7 | 14.6 | 3.9 | 1.5 |
| 1.2–1.3 | 5.0 | 8.3 | | |
| >1.3 | | 8.4 | | |
| Total | 100 | 100 | 100 | 100 |
| Mean overall scale economies | 1.022 | 1.033 | 0.982 | 0.756 |
| Standard deviation | 0.130 | 0.241 | 0.088 | 0.091 |

The largest proportion of overall SE of the NOWCs with low SMR within 1.0–1.1 is 45.1%, and the average overall SE value amounts to 0.982. The largest proportion of the overall SE of NOWC farmers using high SMR within 0.7–0.8 is 63.8%. The overall SE value of the NOWC farmers is below 1. As such, the cultures are in the stage of decreasing return to scale, meaning that any gain in output does not contribute to reducing the average cost.

Table 5 shows the two-factor variation analysis results of the overall SE value of milkfish farmers with different culture characteristics. As for NOWC farming households, polycultures with low SMR have the highest overall SE value at 0.982. In OWCs, the overall SE value is significantly higher.

According to the specific SE estimates, the average values of specific SE of OWC farmers using low or high SMR are 1.023 and 1.005, respectively. As the specific SE value of milkfish is greater than 1, the farmers can slightly increase the milkfish production scale. For OWCs with low SMR, the stocking of 15,000–20,000 fry/ha of milkfish has a higher SE value. Polycultures using high SMR have a higher SE value if the stocking density of milkfish is lower than 15,000 fry/ha. The average specific SE of NOWC farmers with low SMR is 0.996, close to 1, thus approaching the stage of constant return to scale. The average specific SE of polycultures with high SMR is 0.765, which is smaller than 1. Such polycultures are at the stage of a decreasing return to scale, meaning that gains in output of milkfish cannot reduce the average cost. In NOWCs with low SMR, stocking 7000–9000 fry/ha of milkfish has a higher SE value.

When the specific SE value of milkfish equals 1, that is, the farmer's production output has reached the lowest level of average cost, the farmer has obtained the Minimum Efficient Scale (MES). To further discuss the influence of white shrimp–milkfish ratios on production SE values, this study used a robust linear regression model to analyze the influence of milkfish output, white shrimp-milkfish ratio, as well as the interaction term of milkfish output and white shrimp-milkfish ratio on the milkfish production SE value.

According to the parameter estimation result of the regression model shown in Table 6, the parameter of estimated milkfish output of OWC and NOWC systems with low SMR reaches statistical significance. The average white shrimp–milkfish ratio is substituted in the regression model to calculate the optimal milkfish production output when the milkfish SE value equals 1. For OWCs with low SMR, the optimal production output per hectare of milkfish is 21,562 kg; and for NOWCs with low SMR, the optimal production output per hectare of milkfish is 5239 kg.

**Table 5.** Specific scale economies and overall scale economies of the observed milkfish farming household clusters.

| Culture mode | White Shrimp–Milkfish Fry Stocking Density Ratio | Milkfish Specific Scale Economies (PSE) | Specific Scale Economies of Different Milkfish Stocking Density (Fry/ha) | | | Overall Scale Economies (OSE) |
|---|---|---|---|---|---|---|
| | | | <15,000 | 15,000–20,000 | >20,000 | |
| Overwinter (OWC) | Low white shrimp–milkfish ratio | 1.023 (0.042) ** | 1.001 | 1.018 | 0.987 | 1.022 (0.043) ** |
| | High white shrimp–milkfish ratio | 1.005 (0.110) ** | 1.060 | 1.036 | 0.947 | 1.033 (0.125) ** |
| | Total | 1.005 (0.046) ** | 1.104 | 1.041 | 0.984 | 1.033 (0.055) ** |
| | F value | | | 2.65 *,b | | |

| Culture mode | White shrimp–Milkfish Fry Stocking Density Ratio | Milkfish Specific Scale Economies (PSE) | Specific Scale Economies of Different Milkfish Stocking Density (Fry/ha) | | | Overall Scale Economies (OSE) |
|---|---|---|---|---|---|---|
| | | | <7000 | 7000–9000 | >9000 | |
| Non-overwinter (NOWC) | Low white shrimp–milkfish ratio | 0.996 (0.072) ** | 1.005 | 1.007 | 0.943 | 0.982 (0.068) ** |
| | High white shrimp–milkfish ratio | 0.765 (0.079) ** | 0.785 | 0.727 | 0.796 | 0.756 (0.079) ** |
| | Total | 0.818 (0.061) ** | 0.825 | 0.795 | 0.841 | 0.829 (0.061) ** |
| | F value | 76.92 **,a | | 5.47 **,c | | 87.68 **,a |

Notes: Estimates based on the coefficients in Table 2. Standard errors, calculated using the delta method, are in parentheses and *p*-values in brackets. ** Significant at 5%, * at 10%. [a] Conducting a test of null hypothesis that the combination means (expressed in mean vectors) caused by two factors of culture mode and the white shrimp–milkfish fry stocking density ratio (SMR) are not statistically different in PSE and OSE. [b] Conducting a test of null hypothesis that the combination means (expressed in mean vectors) caused by two factors of the white shrimp–milkfish fry stocking density ratio (SMR) and milkfish stocking density are not statistically different in specific scale economies of overwinter culture. [c] Conducting a test of null hypothesis that the combination means (expressed in mean vectors) caused by two factors of the white shrimp–milkfish fry stocking density ratio (SMR) and milkfish stocking density are not statistically different in specific scale economies of non-overwinter culture.

**Table 6.** Estimation result of robust linear regression between milkfish production outputs, white shrimp–milkfish fry stocking density ratio, and milkfish specific scale efficiency of the observed farming household clusters.

| | Low White Shrimp–Milkfish Ratio | | | |
|---|---|---|---|---|
| Variables | Coefficients | Standard Error | *t* | *p* > \|*t*\| |
| $Y_i$ | −0.000008 | 0.000002 | −3.68 | 0.000 |
| Ratio | 0.000203 | 0.001423 | 0.14 | 0.887 |
| $Y_i \times$ Ratio | 0.00000003 | 0.00000008 | 0.33 | 0.741 |
| Constant | 1.1544 | 0.0416 | 27.69 | 0.000 |

| | High white shrimp–milkfish ratio | | | |
|---|---|---|---|---|
| Variables | Coefficients | Standard Error | *t* | *p* > \|*t*\| |
| $Y_i$ | −0.000021 | 0.000115 | −0.02 | 0.986 |
| Ratio | −0.000787 | 0.021047 | −0.04 | 0.970 |
| $Y_i \times$ Ratio | 0.0000001 | 0.000002 | 0.05 | 0.958 |
| Constant | 1.0029 | 1.2775 | 0.79 | 0.437 |

| | Low white shrimp–milkfish ratio | | | |
|---|---|---|---|---|
| Variables | Coefficients | Standard Error | *t* | *p* > \|*t*\| |
| $Y_i$ | −0.000046 | 0.000014 | −3.10 | 0.003 |
| Ratio | −0.007320 | 0.002205 | −3.32 | 0.002 |
| $Y_i \times$ Ratio | 0.00000064 | 0.00000038 | 1.69 | 0.098 |
| Constant | 1.3697 | 0.0838 | 16.33 | 0.000 |

| | High white shrimp–milkfish ratio | | | |
|---|---|---|---|---|
| Variables | Coefficients | Standard Error | *t* | *p* > \|*t*\| |
| $Y_i$ | 0.000013 | 0.000018 | 0.73 | 0.465 |
| Ratio | 0.000372 | 0.000944 | 0.39 | 0.695 |
| $Y_i \times$ Ratio | 0.00000005 | 0.0000002 | −0.35 | 0.728 |
| Constant | 0.6798 | 0.1073 | 6.33 | 0.000 |

As shown in Table 6, the milkfish–white shrimp ratio parameter estimation of NOWCs with low SMR is negative and statistically significant. As such, the SE value decreases when the farmer increases the quantity of white shrimp in the polyculture. The average milkfish output is substituted in the regression model, and the optimal white shrimp–milkfish ratio is 31 (31 white shrimp released for one individual of milkfish) when the milkfish SE value equals 1. The interaction term parameter estimation of milkfish output and white shrimp–milkfish ratio is positive, meaning that the SE value increases as the milkfish output and white shrimp–milkfish ratio decrease. The farmer may thus increase output to reduce the average cost (Figure 2).

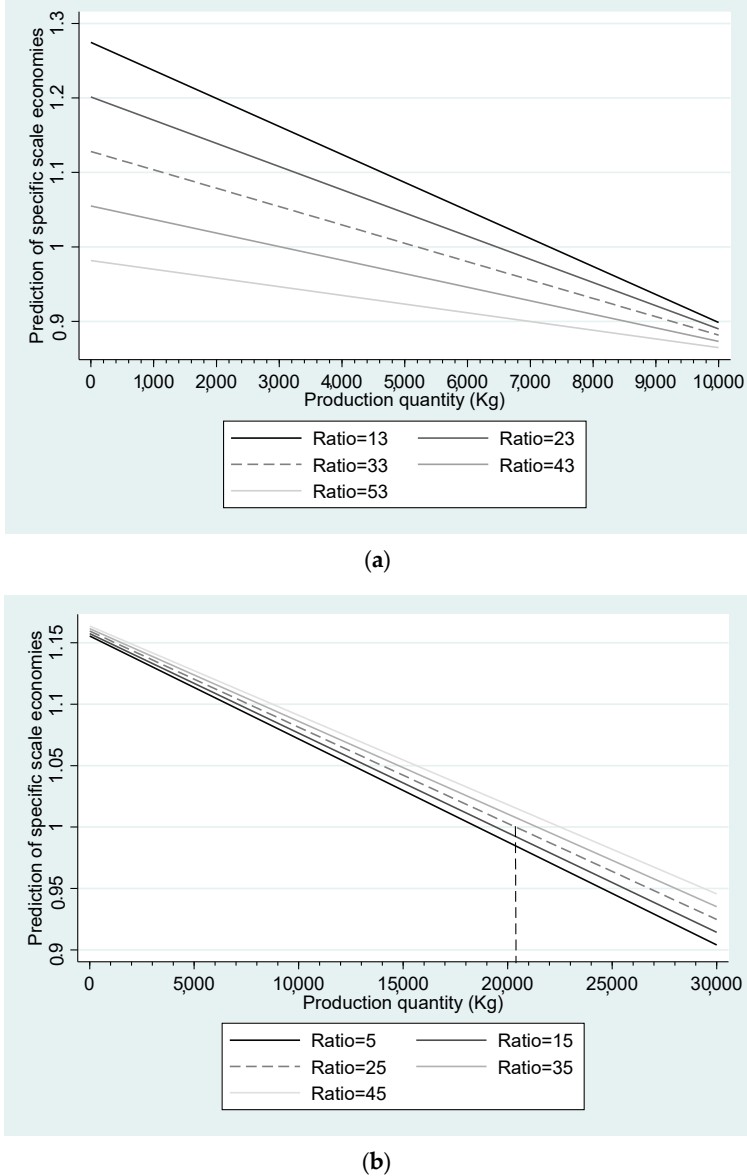

(**a**)

(**b**)

**Figure 2.** The interaction effect of fry stocking density ratio on prediction of specific scale economies in OWC and NOWC systems with low SMR. (**a**) Interaction effect of fry stocking density ratio on the prediction of specific scale economies in NOWCs of milkfish with low SMR. (**b**) Interaction effect of fry stocking density ratio on prediction of specific scale economies in OWCs of milkfish with SMR.

### 3.6. Determinants of Scale Economies

Table 7 shows the regression model parameter estimation result of the factors influencing overall SE values. For farmers using a low SMR, the parameter estimate of fresh water and overwinter culture is positive and statistically significant. This suggests that

the freshwater aquaculture and overwinter culture of milkfish have overall SE, and the farmer can reduce the production cost by increasing the output. The parameter estimates of milkfish stocking density, stocking size, and farmer's age are negative, meaning that low milkfish stocking density, small milkfish fry, and young farmers, exhibit overall SE. For farmers using a high SMR, the parameter estimates of pond depth, overwinter culture, and farmer's age are positive and statistically significant. This indicates that high pond depth, overwinter culture of milkfish and senior farmers have overall SE. The parameter estimates of milkfish stocking density, stocking size, and culture area are negative, meaning that a low milkfish stocking density, small milkfish fry, and small culture area have overall SE.

**Table 7.** Coefficient estimates of overall scale economies' determinant regression for low and high white shrimp–milkfish ratio.

| Parameter | Low White Shrimp–Milkfish Ratio | | High White Shrimp–Milkfish Ratio | |
|---|---|---|---|---|
| | Coefficient | S.E. | Coefficient | S.E. |
| Constant | 1.07201 ** | 0.10418 | 0.74905 ** | 0.06019 |
| Area | 0.00018 | 0.00554 | −0.01080 ** | 0.00182 |
| Time | −0.00924 | 0.00734 | −0.00116 | 0.00398 |
| Depth | −0.00802 | 0.01679 | 0.01844 ** | 0.00799 |
| Fresh | 0.10045 ** | 0.03728 | 0.03557 | 0.02550 |
| $Density_m$ | −0.0000029 ** | 0.0000016 | −0.0000043 ** | 0.0000018 |
| $Density_s$ | −0.00000002 | 0.00000004 | 0.00000001 | 0.00000002 |
| Size | −0.01951 ** | 0.00947 | −0.01171 ** | 0.00428 |
| Age | −0.00266 ** | 0.00097 | 0.00147 ** | 0.00058 |
| Exp | 0.00122 | 0.00144 | 0.00034 | 0.00073 |
| Edu | −0.01674 | 0.03657 | 0.00593 | 0.01688 |
| winter | 0.34013 ** | 0.03371 | 0.29896 ** | 0.01921 |
| $R^2$ | 0.558 | | 0.851 | |
| F-value | 36.59 ** | | 61.49 ** | |

S.E. refer to standard error. ** significance levels are at 5%.

## 4. Discussion

The results of this study differ from previous findings regarding the production efficiency of milkfish polycultures. More specifically, previous research suggests that farmers may reduce the average culture cost by expanding the production scale. This study advises milkfish farmers to modify the white shrimp–milkfish stocking density ratio according to the culture period. That is, OWC farming households of milkfish polycultures have production SE and may thus reduce the average culture cost by expanding the production scale. NOWC famers, on the other hand, fail to have production SE. As such, increases in production scale are unlikely to cause a reduction in the average culture cost.

Moreover, milkfish farmers often increase the quantity of white shrimp in milkfish polycultures in attempts to increase revenues. Although white shrimp help improve the water quality of polycultures, there may be adverse effects if the shrimp stocking density is too high [10,11]. This study finds that NOWC systems have considerably higher SMR than OWCs. More importantly, any reduction in the quantity of white shrimp in NOWCs increases the survival rate of milkfish and white shrimp. As shown in Table 2, the survival rate of milkfish between NOWCs with high and low SMR differs by two percentage points, whereas the survival rate of white shrimp almost doubled (see Table 2). This study suggests that the white shrimp–milkfish ratio is reduced to 31:1.

The own-price elasticities of labor, fry, capital, feed, and other input factors of OWC and NOWC systems are smaller than 1, implying that the input factor use is rigid when the farmer faces input factor price changes. OWCs with high SMR are sensitive to fluctuations in the fry price, since fry is the input factor exhibiting the highest own-price elasticity. Given the extended grow-out period of OWC, farmers may, however, adapt to changes in the fry price by modifying the stocking period and density.

Many studies found substitutional relations between culture input factors. Chiang et al. (2004), for example, revealed substitutional relations between fry and feed, and between fry and other input [13]. This study finds similar relations in OWCs but asserts that substitutability of input factors depends on the used SMR. That is, farmers of OWCs with low SMR may use the input factor other as a substitute for fry; however, farmers of OWCs with high SMR are rather advised to substitute feed for fry to compensate for price hikes.

Furthermore, in a recent study on hard clam cultures, Chang et al. (2020) concludes that fry and high levels of substitutability in cultures with high survival rates [27]. The findings of this study, however, show that the substitutability is conditioned by the culture period. That is, fry and capital of NOWCs exhibit high levels of substitutability, whereas such substitutability could not be established in OWC farming households. Thus, only farmers of NOWCs may benefit from increases in capital equipment input during price hikes.

In terms of factors influencing SE, findings of the study show that low density stocking of milkfish and small milkfish fry are factors that may help farmers achieve SE to reduce costs. Since high milkfish stocking density lengthens the culture time, and large fry increase the fry cost, these factors are unfavorable in terms of reducing costs. Moreover, farmers of polycultures with high SMR may reduce the culture area to achieve SE.

## 5. Conclusions

In this study, a translog cost function model was used to analyze the production scale economy and input demand price elasticity of four milkfish polyculture systems with two different culture periods (OWC and NOWC) and two different white shrimp–milkfish fry stocking ratios (low SMR: 10–55 fry/ha; high SMR: 56–100 fry/ha).

The findings show that the four milkfish polyculture systems require different operational adjustments to increase production while reducing the average culture cost. More specifically, overwinter cultures (OWC) have economies of scale. Farmers may reduce the average cost by expanding the production scale. Non-overwinter polycultures (NOWC) with high SMR are at the stage of decreasing return to scale, meaning that gains in output of milkfish cannot reduce the average cost. In terms of input factor use, farmers of OWCs with high SMR are sensitive to fluctuations in fry price, since fry constitutes the input factor exhibiting the highest own-price elasticity. Moreover, fry and feed of OWCs with high SMR have high levels of substitutability, whereas fry and other input exhibit substitutability in OWCs with low SMR. In NOWCs with high SMR, fry and capital have substitutability. It is thus recommended to modify the input factor use according to the culture mode and the white shrimp–milkfish stocking density ratio.

White shrimp is widely used in polycultures in Taiwan to increase profits. The study found that NOWC polycultures have considerably higher SMR than OWCs. As such, the polycultures exhibit decreasing survival rates which are mainly caused by a deterioration of the water quality in fishponds. Moreover, at such high SMRs, the polyculture system cannot achieve economies of scale. It is thus recommended to reduce the SMR to 31:1 to achieve economies of scale in production and increase the survival rate of milkfish and white shrimp.

The findings may assist milkfish farmers in their attempt to modify the production scale and input factor use for milkfish polycultures with white shrimp to achieve economic efficiency of production.

In terms of research limitations, this study analyzes the economic benefits of aquaculture based on two clusters covering a range of different SMRs. In future research, analysis could focus on smaller clusters of polyculture ratios to understand the characteristics of aquaculture operations with different SMRs.

**Supplementary Materials:** The following are available online at https://www.mdpi.com/article/10.3390/fishes7030110/s1, Table S1: Canonical discriminant functions analysis of four categories based on cost variables; Table S2: Canonical discriminant functions analysis of four categories based on Farm and farmer characteristics; Table S3: Coefficient estimates of translog cost function and cost shares of the observed farming household clusters.

**Author Contributions:** Conceptualization, W.-T.P.; methodology, W.-T.P. and J.-M.L.; software, H.-C.Y.; validation, W.-T.P.; formal analysis, W.-T.P.; investigation, L.-M.H.; resources, Y.-H.L.; data curation, Y.-H.L.; writing—original draft preparation, W.-T.P.; writing—review and editing, W.-T.P., C.S. and J.-M.L.; supervision, L.-M.H.; project administration, C.-Y.Y.; funding acquisition, W.-T.P. All authors have read and agreed to the published version of the manuscript.

**Funding:** This research received no external funding.

**Institutional Review Board Statement:** Not applicable.

**Data Availability Statement:** Data available on request due to restrictions e.g., privacy or ethical.

**Acknowledgments:** The authors are thankful to Yi-Wei Huang for collecting the data of the study.

**Conflicts of Interest:** The authors declare no conflict of interest.

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
