# Peer review of "Effect of Culture Period and Stocking Density on Input Demand and Scale Economies of Milkfish (Chanos chanos) Polycultures with White Shrimp (Penaeus indicus)"

_fishes, doi:10.3390/fishes7030110_

Round 1
Reviewer 1 Report
The paper is dealing with a fish-economics topic with a unique approach. Due to this approach and the éess researched topic the paper is original and gap-filling.
The title is appropriate, covering and reflecting the content well. The abstract is too long but acceptable. Keywords are appropriate.
The introduction is sufficient, highlighting the research goal and context.
However, the literature review is completely missing. This is unacceptable for an article at this scientific level. While there is some literature review in the introduction part, the number analytical depth of processed sources is not appropriate. Usually, a literature review should be organized into a separate chapter where the relevant international and domestic literature is overviewed and analyzed in a critical and comparative way. Also, attention should be paid to involving the latest sources, preferably from WoS or Scopus listed, Q categorized journals.
The methodology toolset is well selected and described detailed. The methods support the analysis and the results.
The limitations of the research are not described.
Author Response
Reviewer1
However, the literature review is completely missing. This is unacceptable for an article at this scientific level. While there is some literature review in the introduction part, the number analytical depth of processed sources is not appropriate. Usually, a literature review should be organized into a separate chapter where the relevant international and domestic literature is overviewed and analyzed in a critical and comparative way. Also, attention should be paid to involving the latest sources, preferably from WoS or Scopus listed, Q categorized journals.
Our response:
Thank you very much for your advice. We have revised the introductory part of the paper. We have omitted a full-fledged literature review in the paper to follow the journal guidelines in terms of the required research manuscript sections. Please accept our apologies.
Reviewer1
The limitations of the research are not described.
Our response:
We have added the limitations under CONCLUSIONS

Reviewer 2 Report
The manuscript is an interesting collaboration, with an observational science approach, which is very important to provide real information of aquaculture activities in different regions of the world. The manuscript used “translog cost function modeling” to assess the economic viability of fish-shrimp polycultures in Taiwan. This empirical model has already been widely used in agricultural sciences around the world, but it is not widely applied in aquaculture. However, there are many results that need discussion. Statistical analyzes need to be revised. All results converge to a single conclusion: reduce the SMR to 31:1 to achieve economies of scale in production. Please find below my suggestions.
Lines 46-67: First three paragraphs: Excessive information about Chanos chanos cultivation in Taiwan. However, there are no bibliographic references, which need to be included. Suggestion: the authors rewrite this early part of the Introduction, focusing on the polyculture of milkfish and white shrimp.
Lines 65-67: Very short paragraph with excess bibliographic references
Lines 91-94: Authors should be more concise in objectives. In my opinion, the two research questions overlap.
Lines 95-97: Use this information in conclusions, but not at the end of the Introduction
Lines 102-103: Suggestion: Change T2 to OWC. Invert the sentences. First mention T1 (NOWC) and then T2 (OWC).
Lines 103-104: Suggestion: Change T1 to NOWC in every manuscript
Lines 110-112: Is there a paper or report that demonstrates this representativeness of the production of the counties of Tainan and Kaohsiung that can be cited here?
Lines 114-115: Include map with Taiwan in a global context.
Lines 116-129: How many fish farms and how many ponds were analyzed during the observational study in the two farming systems? What differentiates the two systems is just the cultivation time? T1 longer growing period and T2 shorter growing period? How much is this difference in terms of days or months?
Lines 139-141: Results
Lines 142-167: The Total Operating Cost of production corresponds to the
Effective Operating Cost plus depreciation and bank interest rate. Are loan interest rates included in the Total Operating Cost calculation?
Lines 217-260: Basic information about the experimental design (such as the quantitative of fish farmers) should be informed at the beginning of the M&M, more specifically in “Culture Mode”.
Lines 262-269: This information should also be at the beginning of M&M in “Culture Mode”. "Stocking density ratio <56": I suggest entering a range. Example: 22 – 56; 57 - 70.....
Lines 271-273: Authors should make it clear at the beginning of the M&M and also in the objectives of the manuscript that four different culture systems will be analyzed and not two systems.
Line 279 (Table 2): Are the data averages? It is very important to present the standard deviations for each variable in the table, if the table is to be kept in the manuscript. An exploratory analysis, such as a principal component analyze (PCA) with all samples or with average values, would be more visually interesting than this table 2.
Lines 282-305: Did the authors choose to work with the mean of the analyzed variables? If this was the option, it is essential to present the standard deviations.
The authors could also apply a parametric or non-parametric analysis of variance, as it is possible that despite the average being different, significant differences are not found between the 4 cropping systems analyzed. A more interesting statistical analysis for this dataset would be an exploratory analysis (PCA, for example), to identify patterns in aquaculture farms.
Line 323 (Table 3): Supplementary material (attachments)
Lines 354-357 (Table 4): Table 4 (354-357). What statistical analysis was applied to verify significant differences between the 4 cropping systems? Did the primary dataset meet the requirements of the analysis used? If not, again the suggestion is an exploratory statistical analysis as a PCA to check for patterns.
Lines 370-372 (Table 6): same comment in table 4
Line 437: The discussion uses subjective terms: slightly expanding the production scale
Lines 447-449: What motivates this statement, if the authors did not carry out an analysis of the water quality of production systems? Not even a bibliographic reference was cited here.
Lines 450-451: Based on what results do the authors make this claim? Would these survival rates be significantly higher?
Lines 479-481: Could it be a spurious correlation?
Lines 486-487: Not related to research. Something obvious.
Lines 509-510: Is this recommendation based on economics or is it considering survival aspects and water quality improvements? All results converge to this conclusion. It needs to be discussed further.
Author Response
Reviewer 2
Lines 46-67: First three paragraphs: Excessive information about Chanos chanos cultivation in Taiwan. However, there are no bibliographic references, which need to be included. Suggestion: the authors rewrite this early part of the Introduction, focusing on the polyculture of milkfish and white shrimp.
Our response:
We have rewritten the Introduction.
Reviewer 2
Lines 65-67: Very short paragraph with excess bibliographic references
Our response:
We have revised that section of the paper.
Reviewer 2
Lines 91-94: Authors should be more concise in objectives. In my opinion, the two research questions overlap.
Our response:
We have revised that section of the paper. (lines 83-92)
Reviewer 2
Lines 95-97: Use this information in conclusions, but not at the end of the Introduction
Our response:
We have revised that section of the paper.
Reviewer 2
Lines 102-103: Suggestion: Change T2 to OWC. Invert the sentences. First mention T1 (NOWC) and then T2 (OWC).
Our response:
We have revised that the paper accordingly.
Reviewer 2
Lines 103-104: Suggestion: Change T1 to NOWC in every manuscript
Our response:
We have made those changes. (lines 98-105)
Reviewer 2
Lines 110-112: Is there a paper or report that demonstrates this representativeness of the production of the counties of Tainan and Kaohsiung that can be cited here?
Our response:
We have added a source. (line 109)
Reviewer 2
Lines 114-115: Include map with Taiwan in a global context.
Our response:
We have revised the map. (line 112)
Reviewer 2
Lines 116-129: How many fish farms and how many ponds were analyzed during the observational study in the two farming systems? What differentiates the two systems is just the cultivation time? T1 longer growing period and T2 shorter growing period? How much is this difference in terms of days or months?
Our response:
We have added the required information in the text (lines 96-112).
Reviewer 2
Lines 139-141: Results
Our response:
We have revised the text accordingly.
Reviewer 2
Lines 142-167: The Total Operating Cost of production corresponds to the
Our response:
We have revised the text accordingly. (line 154)
Reviewer 2
Effective Operating Cost plus depreciation and bank interest rate. Are loan interest rates included in the Total Operating Cost calculation?
Our response:
Since there are very few households that require a bank loan, interests payments are included in other costs.
Reviewer 2
Lines 217-260: Basic information about the experimental design (such as the quantitative of fish farmers) should be informed at the beginning of the M&M, more specifically in “Culture Mode”.
Our response:
We have revised the paper accordingly. (lines 109-112)
Reviewer 2
Lines 262-269: This information should also be at the beginning of M&M in “Culture Mode”. "Stocking density ratio <56": I suggest entering a range. Example: 22 – 56; 57 - 70.....
Our response:
We have revised the paper accordingly. (lines 264-272)
Reviewer 2
Lines 271-273: Authors should make it clear at the beginning of the M&M and also in the objectives of the manuscript that four different culture systems will be analyzed and not two systems.
Our response:
We have revised the paper accordingly. (lines 83-91)
Reviewer 2
Line 279 (Table 2): Are the data averages? It is very important to present the standard deviations for each variable in the table, if the table is to be kept in the manuscript. An exploratory analysis, such as a principal component analyze (PCA) with all samples or with average values, would be more visually interesting than this table 2.
Our response:
We have revised the paper accordingly.
We have also performed a canonical discriminant function analysis (lines 310-321)
Reviewer 2
Lines 282-305: Did the authors choose to work with the mean of the analyzed variables? If this was the option, it is essential to present the standard deviations.
The authors could also apply a parametric or non-parametric analysis of variance, as it is possible that despite the average being different, significant differences are not found between the 4 cropping systems analyzed. A more interesting statistical analysis for this dataset would be an exploratory analysis (PCA, for example), to identify patterns in aquaculture farms.
Our response:
See above.
Reviewer 2
Line 323 (Table 3): Supplementary material (attachments)
Our response:
We revised that. (line 330)
Reviewer 2
Lines 354-357 (Table 4): Table 4 (354-357). What statistical analysis was applied to verify significant differences between the 4 cropping systems? Did the primary dataset meet the requirements of the analysis used? If not, again the suggestion is an exploratory statistical analysis as a PCA to check for patterns.
Our response:
The input factor elasticity estimation results shown in Table 4 (now Table 3) base on methods applied in similar studies discussed in the literature. That is, the parameter estimates and mean variable values are used in the calculation of input factor elasticities and scale economies.
Reviewer 2
Lines 370-372 (Table 6): same comment in table 4
Our response:
A series of test of null hypothesis were carried out to verify the differences (see lines 385-393).
Reviewer 2
Line 437: The discussion uses subjective terms: slightly expanding the production scale
Our response:
We have revised that part. (line 461)
Reviewer 2
Lines 447-449: What motivates this statement, if the authors did not carry out an analysis of the water quality of production systems? Not even a bibliographic reference was cited here.
Our response:
We have added sources. (line 471)
Reviewer 2
Lines 450-451: Based on what results do the authors make this claim? Would these survival rates be significantly higher?
Our response:
We have revised that paragraph (lines 472-476).
Reviewer 2
Lines 479-481: Could it be a spurious correlation?
Our response:
Since we cannot rule out that the result is spurious we decided to delete that finding from our paper.
Reviewer 2
Lines 486-487: Not related to research. Something obvious.
Our response:
We have deleted that sentence.
Reviewer 2
Lines 509-510: Is this recommendation based on economics or is it considering survival aspects and water quality improvements? All results converge to this conclusion. It needs to be discussed further.
Our response:
We have revised that paragraph (lines 519-525)

Round 2
Reviewer 2 Report
Most changes were made by the authors. Even not having carried out all the suggested changes, such as the application of multivariate analysis to the data set, aiming to identify possible patterns between the production systems, I consider that the manuscript has the quality for publication.